

# Modelling 3D permeability distribution in alluvial fans using facies architecture and
# geophysical acquisitions
Lin Zhu [1], Huili Gong [1], Zhenxue Dai [2], Gaoxuan Guo [3], Pietro Teatini [4]
[1]College of Resource Environment and Tourism, Capital Normal University, Laboratory Cultivation Base
of Environment Process and Digital Simulation, Beijing, China
[2]Earth and Environmental Sciences Division, Los Alamos National Laboratory, Los Alamos, New
Mexico, United States
[3] Beijing Institute of Hydrogeology and Engineering Geology, Beijing, China
[4] Department of Civil, Environmental and Architectural Engineering, University of Padova, Italy
*Correspondence to*: Lin Zhu hi-zhulin@163.com; Huili Gong gonghl@263.com
**Abstract**. Alluvial fans are highly heterogeneous due to complex depositional processes, which make
difficult to characterize the spatial distribution of the hydraulic conductivity $K$. An original methodology
is developed to identify the spatial statistical parameters (mean, variance, correlation range) of the
hydraulic conductivity in a three-dimensional setting by using geological and geophysical data. The
Chaobai River alluvial fan in the Beijing Plain, China, is used as an example to test the proposed approach.
Due to the non-stationary property of the $K$ distribution in the alluvial fan, a multi-zone parameterization
approach is applied to analyze the conductivity statistical properties of different hydrofacies in the various
zones. The composite variance in each zone is computed to describe the evolution of the conductivity
along the flow direction. Consistently with the scales of the sedimentary transport energy, the results show
that conductivity variances of fine sand, medium-coarse sand, and gravel decrease from the upper (Zone
1) to the lower (Zone 3) portion along the flow direction. In Zone 1, sediments were moved by higher-
energy flooding, which induces bad sorting and larger conductivity variances. The composite variance
confirms this feature with statistically different facies from Zone 1 to Zone 3. The results of this study
provide insights to improve our understanding on conductivity heterogeneity and a method for
characterizing the spatial distribution of $K$ in alluvial fans.



## 1 Introduction

Alluvial fans usually house valuable groundwater resources because of significant water storage and
favorable recharge conditions. Sedimentary processes forming alluvial fans are responsible for their
complex long-term evolution. Usually, the coarsest material (gravel) is deposited in the upper fan, with
the gravel passing into sand in the middle of the fan and then into silt and clay in the tail. A high
heterogeneity characterizes the deposit distribution because of the shifting over time of the sediment-
transporting streams (Zappa et al., 2006).
Conductivity distributions in alluvial fans can be assigned according to the various hydrofacies simulated
by conditional indicator geostatistical methods (Eggleston and Rojstaczer 1998; Ritzi et al., 2000, 2004;
Proce et al., 2004; Dai et al., 2005; Harp et al., 2008; Hinnell et al., 2010; Soltanian et al., 2015; Zhu et
al., 2015a). However, the geostatistical methods require the stationary assumption, i.e. the distribution of
the volumetric proportions and correlation lengths of hydrofacies converge to their mean values in the
simulation domain. The hydrofacies and hydraulic conductivity ($K$) distributions in alluvial fans are
generally non-stationary (Weissmann and Fogg, 1999; Anderson, 2007; Zhu et al., 2016a). Hence, the
use of these methods may cause large characterization errors and add significant uncertainty to the
predictions achieved by groundwater flow and contaminant transport models (Eggleston and Rojstaczer
1998; Irving and Singha 2010; Dai et al., 2014a). Zhu et al., (2016a) adopted a local-stationary assumption
by dividing the alluvial fan into three zones along the flow direction of the Chaobai River, China. The
zones were properly detected based on the statistical facies distribution. Then, the indicator simulation
method was applied to each zone and the simulated hydrofacies distribution in the three zones was used
to guide modelling the $K$ distribution.





Hydraulic conductivity of granular deposits generally varies with grain size, porosity, and sorting.
Traditional methods for $K$ estimate, e.g. well test, permeameter measurements, and grain-size analyses
(Niwas et al., 2011), are very expensive, time-consuming, and make difficult to provide representative
and sufficient field data for addressing spatial variations of conductivity. Recently, data fusion techniques
have been developed for coupled inversion of multi-source data to estimate $K$ distributions for
groundwater numerical modeling. Geophysical data (such as surface electric resistivity and various
logging data) are relatively inexpensive and can provide considerable information for characterizing
subsurface heterogeneous properties (Hubbard et al., 2001; Yeh et al., 2002; Dai et al., 2004a; Morin
2006; Sikandar et al., 2010; Bevington et al., 2016). Electric resistivity data have been proven useful to
derive sediment porosity distributions (Niwas and Singhal 1985; Niwas et al., 2011; Niwas and Celik
2012; Zhu et al., 2016b).
This study proposes an integrated approach to reconstruct the three-dimensional configuration of
conductivity in alluvial fans by combining the hydrofacies spatial heterogeneity provided by a multi-zone
transition probability model with hydrogeological and hydrogeophysical measurements, in particular
resistivity loggings and electrical soundings.  We assume the $K$ distributions are local-stationary, i.e. the
mean and variance of log conductivity are convergent in each hydrofacies and in each local zone.
Therefore, we can compute the $\log_{10}(K)$ semivariogram in each hydrofacies and in each zone The Chaobai
alluvial fan in the northern Beijing Plain, China, was selected as study area to test the proposed integrated
approach.




## 2 Material and Methods

### 2.1 Study area

The study area belongs to the Chaobai River alluvial fan, in the northern Beijing Plain (northern latitude $40°-40°30'$, eastern longitude $116°30'-117°$), with an area of 1,150 km$^2$ (Fig. 1a). The Chaobai River is the second largest river flowing through the Beijing Plain from north to south. The ground elevation decreases southward with an average 2‰ slope. Quaternary sediments were mainly deposited by flooding events with turbulent flow and consist of porous strata containing groundwater. The aquifer system in the alluvial fan can be divided into three zones according to the lithological features (Fig. 1): an upper fan zone (or Zone 1) with coarse sediments (e.g., sandy-gravel aquifers), a middle upper fan zone (or Zone 2) where medium-coarse sediments (e.g., sandy-gravel to sandy-silt aquifers) were laid down , and a fine-sediment (e.g., sand and clay multiple aquifers) middle-lower fan zone (or Zone 3). Four hydrofacies, including sub-clay and clay (C), fine sand (FS), medium-coarse sand (MS), and gravel (G), were classified based on the interpretations of the cores and textural description of almost 700 boreholes (Zhu et al., 2015).

The study area is one of the most important regions for the supply of groundwater resource to Beijing. The Huairou emergency groundwater resource region (hereafter EGRR) with an area of 54 Km$^2$ is located in Zone 1. The total groundwater withdrawal amounted to $1.2×10^8$ m$^3$ in 2003. Several well-fields belonging to the so-called "water supply factory" were drilled along the Chaobai River in Zone 1 and the upper Zone 2. Most of these well-fields were built in 1979 with a designed groundwater pumping volume of $1.6×10^8$ m$^3$ per year. The average thickness of the exploited aquifer system is approximately 300 m.



The long-term over-exploration of the aquifer system has resulted in a serious drawdown of water levels,
which has reduced the exploitable groundwater resources and induced geological disasters, mainly land
subsidence, fault reactivation, and ground fissures (Cheng et al., 2015; Yang et al., 2015; Zhu et al., 2015).
In 2010, the annual groundwater withdrawal at the EGRR and the water factory decreased to $0.86\times10^8$
$m^3$ and $0.65\times10^8$ $m^3$, respectively.
The largest cumulative land subsidence from June 2003 to January 2010 was quantified in approximately
340 mm by Zhu et al., (2013, 2015) in Tianzhu County to the south. The characterization of the
distribution and spatial variability of the hydraulic conductivity is vital for an optimal use of the limited
water resources in this area.
**2.2 Methodological approach**
Nowadays, a large set of hydraulic conductivity samples can be derived by integrating through appropriate
relations of various geological data, including hydrogeophysical measurements, borehole
lithostratigraphies, and hydrogeological information (total dissolved solid TDS and groundwater level).
These databases can be statistically processed to derive the spatial variation of $\log_{10}(K)$ for various facies,
including clay, fine sand, medium-coarse sand, and gravel.
In this paper, the statistical assessment is separately carried out for separated zones, building-up
experimental semivariograms that are fitted with exponential models. The optimal parameters of these
latter are estimated through a generalized output least squares (OLS) criterion. Then, the composite
semivariograms are computed using a hierarchical sedimentary architecture (Ritzi et al., 2004; Dai et al.,
2005) to obtain the $K$ variance in each zone. Finally, the configuration of $\log_{10}(K)$ is simulated through a



multiple-zone sequential Gaussian algorithm with estimated statistic parameters reflecting the *K* spatial
structures in the alluvial fan. Figure 2 shows the steps involved in the developed approach.
**2.3 Data set**
**2.3.1   Geophysical data**
Geophysical data include resistivity loggings and vertical electrical soundings. There are six well-electric
logs continuously recording the formation resistivity versus depth. Five logs were collected in Zone 2 and
one in Zone 3. Each well log has a lithological description, which helps to relate the resistivity values to
the corresponding facies.
The average resistivity of G is the largest, with a value of 198 Ω m, and that of C is the smallest with a
value of 24 Ω m. Figure 3 compares the outcome of logging data in term of resistivity versus depth and
the corresponding stratigraphy, where the groundwater depth is 12 m. The log was acquired in the eastern
part of Zone 2. The average resistivity from 32.4 m to 40.5 m depth, where the sediments are mainly G
and MS, is 70.8 Ω m. The resistivity curve shows two evident peaks from 97 m to 102 m and between 81
m and 84.5 m depth, where the MS is located.
The C resistivity is relatively low due to the good intrinsic electrical conductivity of this facies. For
example from 16.5 m to 23.5 m depth, where C is the prevalent facies, a low resistivity equal to 27.2 Ω
m is recorded. Since a hydrofacies with a smaller grain size has a greater total surface area, the resistivity
difference can partially reflect the distributions of particle sizes and the hydrofacies composition.
Vertical electrical soundings (VES) using the Schlumberger electrode configuration were carried out by
the Beijing Institute of Hydrogeology and Engineering Geology (BIHEG). A number of 113 detecting





positions were selected, with a maximum half current electrode space equal to 340 m and the potential
electrode space ranging from 1 to 30 m. All the sounding data (1356 VES measurements) recorded the
apparent resistivity of the porous medium. These data were inverted to real resistivity using the nonlinear
Occam inversion method (Constable et al., 1987), with a low root mean square relative error of 2%.
**2.3.2  Geological and hydrogeological data**
Almost 700 borehole lithostratigraphies were collected in the study area. The sedimentary deposits show
large heterogeneity from the upper to the lower fan zone. In Zone 1, the dominant facies is G with a
volumetric proportion of 53%. The volumetric proportion of C is 16%. In Zone 2, the volumetric
proportion of C increases to 40%, while that of G decreases sharply to 24%. In Zone 3, the proportion of
G decreases further to 6% and that of C increases to 50% (Table 1). More detailed information is given
in Zhu et al., (2016a). The lithological information in a buffer zone of 200 m around the VES locations
has been used to represent the actual facies distribution in the surrounding of the geophysical acquisitions.
A number of 35 hydrochemistry measurements with a depth from 20 m to 270 m were obtained throughout
the area. The minimum, maximum and average TDS values are 423 mg/l at the depth of 180 m, 943 mg/l
at the depth of 50 m, and 692 mg/l, respectively. Generally, the TDS is very low with the higher values
measured in the south-western part of the study area. Because of the relatively small dataset and the
observed low variability, in this paper the TDS variation in the vertical direction has been neglected. A
TDS map was obtained by interpolating the available records using an Ordinary Kriging method with a
spherical semivariogram model.



A large number of depth of water level measurements were also collected to map the thickness of the
unsaturated unit. The TDS and groundwater level at each VES and resistivity log location were derived
from the interpolated surfaces.

### 2.3.3 Hydraulic conductivity estimates from geophysical acquisitions

The hydraulic conductivity $K$ was estimated using the Kozeny-Carman equation:
$$K(x, y, z) = \frac{\delta g}{\mu} \times \frac{d_{(x,y,z)}^2}{180} \frac{\phi_{(x,y,z)}^3}{(1-\phi_{(x,y,z)})^2} \qquad (1)$$

which is widely accepted to derive the hydraulic conductivity from grain size and porosity (Soupious et
al., 2007; Utom et al., 2013; Khalil et al., 2013; Zhu et al., 2016). In Eq. (1), $d_{(x,y,z)}$ is the representative
grain diameter (mm) at location $(x,y,z)$, which was determined according to the lithology information, $g$
is gravity, $\mu$ the kinematic viscosity (kg/(m·s)), $\delta$ the fluid density, and $\phi_{(x,y,z)}$ the porosity. $\phi$ was
estimated using Archie's law (Eq. (2)), which relates the bulk resistivity of granular medium to porosity:
$$\rho = \alpha \rho_w \phi^{-m} s_w^{-n} \qquad (2)$$

where $\rho$ is the saturated formation resistivity ($\Omega$ m), $\alpha$ the pore-geometry coefficient associated with the
medium ($0.5 \leq \alpha \leq 2.5$), $m$ the cementation factor ($1.3 \leq m \leq 2.5$) (Massoud et al., 2010; Khalil and Santos
2013), $s_w$ the water saturation, and $n$ the saturation index. The pore fluid resistivity ($\Omega$ m) $\rho_w$ is calculated
using the following experimental relation:
$$\rho_w = \frac{5.6(\text{TDS})^b}{1+\beta(t-18)} \qquad (3)$$

with TDS in (g/L), temperature $t$ in (°C), $b$ and $\beta$ being constant parameters (Wu et al., 2003). For the
most common electrolytes, $b = -0.95$ and $\beta = 0.025$.



The logarithmically transformed values of the estimated hydraulic conductivity ($\log_{10}(K)$) were used for
the geostatistical analysis because of its normal distribution (Neuman, 1990). There are 102, 2077, and
1716 conductivity samples in Zone 1, Zone 2, and Zone 3, respectively. Considering that Archie's law
can only be used for clay-free granular sediments, the $K$ values of C were not estimated in this study.
Based on available information, it has been reasonably assumed that clay fraction is negligible in G, MS,
and FS facies. The statistics of $\log_{10}(K)$ for the three facies in three zones are listed in Table 2. The mean
$\log_{10}(K)$ values decrease from Zone 1 to Zone 3, consistently with the sedimentary transport processes in
the alluvial fan. In the upper region (Zone 1), high water flowing energy made the deposits consisted
mainly of larger-grained particles and the coarse-grained sediments are dominant. In the southern part
(Zone 3), the deposits change to relatively fine-grained particles. The mean $\log_{10}(K)$ of gravel is greater
than 2.4 (log(m/d)) and that of fine sand is less than 0.2 (log(m/d)). The lithological information at the
depth of the conductivity samples shows that volumetric proportions of FS and MS increase and that of
G decreases from Zone 1 to Zone 3. The results are consistent with the statistic outputs deduced from 694
borehole data by Zhu et al., (2016a).
**2.4 Statistical Methods**
**2.4.1    Semivariogram of hydraulic conductivity**
Semivariogram describes the degree of spatial dependence of a spatial random field or stochastic process.
It is a concise and unbiased characterization of the spatial structure of regionalized variables, which is
important in Kriging interpolations and conditional simulations. The experimental semivariogram:
$$\hat{r}_k\left(h_\varphi\right) = \frac{1}{2N(h)}\sum_{(o,p)\in N(h)}(Y(z_o) - Y(z_p))^2 \tag{4}$$





can be fitted by an exponential model (e.g., Dai et al., 2014b):

$$r_k\left(h_\varphi\right) = \sigma^2(1 - e^{\frac{-3h}{\lambda}}) \qquad (5)$$

where $\hat{r}_k\left(h_\varphi\right)$ and $r_k\left(h_\varphi\right)$ are the experimental and model semivarograms of log conductivity $Y$ for the
$k^{th}$ facies at a lag distance $h$ along the $\varphi$ direction. In this paper we calculate the semivarograms in the
vertical and dip directions. $N(h)$ is the number of pair measuring points $z_o$ and $z_p$ separated by a $h$ lag
distance, $\sigma^2$ is the variance, and $\lambda$ the correlation range.
The variance and range were optimized using the least-squares criterion, which was solved by the
modified Gauss-Newton-Levenberg-Marquardt method (Clifton and Neuman, 1982; Dai et al., 2012).
The sensitivity equation method was derived to compute the Jacobian matrix for iteratively solving the
gradient-based optimization problem (Samper and Neuman 1986; Carrera and Neuman 1986; Dai and
Samper, 2004; Samper et al., 2006; Yang et al., 2014; Zhu et al., 2016a). The two sensitivity coefficients
$\frac{\partial r_k}{\partial \sigma^2}$ and $\frac{\partial r_k}{\partial \lambda}$ are the partial derivatives of the semivariogram with respect to variance and range:

$$\frac{\partial r_k}{\partial \sigma^2} = 1 - e^{\frac{-3h}{\lambda}} \qquad (6)$$

$$\frac{\partial r_k}{\partial \lambda} = -\sigma^2 \cdot 3h \cdot e^{\frac{-3h}{\lambda}} \cdot \lambda^{-2} \qquad (7)$$


### 2.4.2   Composite semivariogram of log conductivity

Once the facies semivariograms were obtained in each zone, the composite semivariogram $\gamma(h)$ could be
calculated through the following equation (e.g., Ritzi et al., 2004):





$$\gamma(h_\varphi) = \sum_{k=1}^{M} \sum_{i=1}^{M} r_{ki}(h_\varphi) p_k t_{ki}(h_\varphi) \tag{8}$$

where $p_k$ and $t_{ki}(h_\varphi)$ are the volumetric proportion of facies $k$ and the transition probability from facies
$k$ to facies $i$ in the $\varphi$ direction with a $h$ lag distance, respectively. Equation 8 delineates the composite
semivarigoram with respect to the individual facies semivariogram and transition probability. The general
shape function and range of the composite semivarigoram can be obtained from individual facies mean
length and volumetric proportion with the methods described in Dai et al., (2005).
The transition probability $t_{ki}(h_\varphi)$ has an analytical solution as derived by Dai et al., (2007):
$$t_{ki}(h_\varphi) = p_k + (\delta_{ki} - p_k) \cdot \exp\left(\frac{h_\varphi}{\lambda_\varphi}\right) \tag{9}$$

where $\delta_{ki}$ is the Kronecker delta and $\lambda_\varphi$ is the integral scale in the direction of $\varphi$. A geostatistical modeling
tool GEOST (Dai et al., 2014b) modified from the Geostatistical Software Library (Deutsch and Journel,
1992) and TPROGS (Carle and Fogg, 1997) was employed to compute the sample transition probabilities
in each zone. The parameters $p_k$ and $\lambda_\varphi$ were optimally estimated through a modified Gauss-Newton-
Levenberg-Marquardt method. More details are provided by Zhu et al., (2016a). The composite
semivariograms for different zones can help us to understand the heterogeneity variations from the upper
to lower part of the alluvial fan, as well as the stationary property (local versus regional) of the facies and
hydraulic conductivity distributions.
**2.4.3  Sequential Gaussian simulation**
The Sequential Gaussian simulation (SGSIM) is a widely used stochastic simulation method to create
numerical model of continuous variables based on the Gaussian probability density function. The process



is assumed to be a stationary and ergodic random process (Deutsch and Journel, 1992; Dimitrakopoulos
and Luo, 2004). This method can preserve the variance and correlation range observed in spatial samples.
SGSIM provides a standardized normal continuous distribution of the simulated variable.
With the assumption that the log conductivity distributions are stationary within each zone, we used
SGSIM simulator implemented into GEOST to model the $\log_{10}(K)$ continuous configuration under a
multiple-zone framework. The conductivity of the FS, MS, and G facies in each zone was simulated
sequentially using the structure characteristics of the semivariograms.
Finally, the three-dimensional conductivity configuration was derived by combining the stochastic
simulated facies (Zhu et al., 2016a) with the SGSIM conductivity distribution and the mean $\log_{10}(K)$ of
the various facies in each zone (Table 2). In detail, since each cell is characterized by specific facies and
zone indices, its conductivity was assigned using the corresponding (in relation to the facies and the zone)
3D SGSIM outcome in that position. Since sub-clay and clay are generally characterized by a low
hydraulic conductivity value, a uniform $K$ value equal to 0.0001 m/d was set to all the C cells.
**3 Results and Discussion**
**3.1 Variation of $\log_{10}(K)$ for the various facies**
The optimized vertical correlation range and variance of the log conductivity semivariogram (Eq. 5) are
listed in Table 3, along with their 95% confidence intervals. The fitting between the experimental and the
model semivariograms is the best in Zone 2 because of the abundant samples, while the fitting in Zone 1
is the worst (Fig. 4). The fitting result of the semivariogram for the G facies is the worst in Zone 1. Two
are the reasons: the first is the high variance of gravel in this zone; the other is the limited number of





samples (102 samples), which makes quite small the pair numbers within each lag spacing. Hence, the
computed semivariogram is highly uncertainty.
The variance of FS, MS, and G in the vertical direction decreases from Zone 1 to Zone 3. In the upper
alluvial fan, sediments were deposited under multiple water flowing events and with bad sorting. The
deposits consist of wide ranges of sediment categories and grain sizes. The variance of G is larger than
1.5, which reflects the high heterogeneity in coarse deposits. The variances of FS and MS are smaller
with values equal to 0.23 and 0.32, respectively. In Zone 3, these values decrease to 0.05 and 0.13,
respectively, with that of G sharply decreasing to 0.62. In the middle-lower fan zone, the conductivity
variation within each facies reduces gradually because the ground surface slope becomes smaller or flat,
the sediment transport energy decreases, and the deposits within the three facies have a good sorting.
Note that the ranges are correlated with the facies structure parameters such as the indicator correlation
scale, mean thickness (or length), and volumetric proportion (Dai et al., 2004b; 2007). The estimated
correlation ranges of FS, MS and G along the vertical direction in Zone 1 do not show big difference with
values equal to 6.0 m, 8.0 m and 6.5 m, respectively. In Zone 2, the ranges of three facies change a lot.
The range of G is almost five and two times that of FS and MS, respectively. Conversely, the range
difference among the facies decreases sharply in Zone 3. The range of G is alike to that of FS with a value
of about 6.0 m, similarly as in Zone 1, and twice as much that of MF. The variation of the structure
parameters of three facies causes the large changes of the correlation ranges from Zone 1 to Zone 3.
Due to the small number of conductivity samples in Zone 1, the variance of $\log_{10}(K)$ along the dip
direction is calculated only in Zone 2 and Zone 3 (Table 4). The variance of G, MS and FS in Zone 2 is



higher than that in Zone 3, as observed along vertical direction. In Zone 2, the variance of FS, MS and G
in the dip direction is gently larger than that along the vertical direction. This occurrence possibly reflects
multiple flooding events that caused particles deposited along the dip direction more heterogeneously
than in the vertical direction during the formation process of Zone 2. Conversely, the variance associated
FS, MS and G is smaller along the dip direction than the vertical one in Zone 3.
**3.2 Composited semivariogram of $\log_{10}(K)$**
The composite semivariogram in the vertical direction at each zone is calculated by Eq. (8), using the
volume proportion (Table 1) and transition probability (Eq. (9)) with the same values of the lag distance
used to compute the facies semivariograms. The values of the optimized variance are 0.68, 0.11, and 0.03
in Zone 1, Zone 2, and Zone 3, respectively. The high flow energy and the large number of flooding
events contributing to sediment deposition are the main causes of the high heterogeneity (largest variance)
of the deposits in the upper part of the alluvial fan. The changes of variance between the three zones
support the utilization of the local-stationary assumption and simulation of multiple-zone based
conductivity distributions for the Chaobai alluvial fan.
**3.3 Configuration of $\log_{10}(K)$**
The configuration of $\log_{10}(K)$ in three dimensions is showed in Fig.7. The distribution of conductivity is
generally consistent with that of the facies. The conductivity of large grain-size sediments is generally
larger, thus on the average $K$ is much larger in the upper zone than in the lower part of the alluvial fan.
The regions with high conductivity (red color in Fig. 7) in Zone 1 are more continuous than that in other
parts. The adjacent cells with the smallest conductivity (blue color in Fig. 7) are obviously located mainly



in Zone 3. The mean conductivity is smaller in the southern part of the study area, where the piezometric
drawdowns in the multi-layer aquifer system were larger and the surface subsidence more serious (Zhu
et al., 2013, 2015).
Based on the three dimensional $K$ configuration, the average value of $K$ in the depth range from 0 m to
300 m amounts to 194 m/d, 25 m/d and 4 m/d in Zone 1, Zone 2, and Zone 3, respectively. These values
are comparable with those provided by the Beijing Institute of Hydrogeology and Engineering Geology
(2007) based on a number of pumping tests carried out over several years in the study area. In this BIHEG
report the average value of $K$ is >300 m/d in Zone 1, between 30 and 100 m/d in Zone 2, and <30 m/d
in Zone 3 (Fig. 1b). The fact that our average $K$ values is gently smaller than these latter is likely due to
the fact that the outcome of pumping tests are generally more representative of coarser sediments.
Conversely, those estimated from the stochastic framework represent more properly the heterogeneous
distributions of the hydraulic conductivity (Zhu et al., 2016b).
Investigating the stochastic results along the vertical direction, it is interesting to notice that the average
$K$ in deep units of Zone 1 and Zone 2 is smaller than that in the shallow strata. For example, in Zone 1
the average $K$ for the cells from 0 m to 100 m deep is 295 m/d, which is three times as much the value for
the depth range between 200 m and 300 m. Conversely, no significant variation of $K$ versus depth is
observed in Zone 3, with only a small decrease of the average $K$ from the deeper to the shallower units.
**4 Conclusions**
This paper proposes a geostatistical method under a multiple zone framework, properly supported by a
large number of geophysical investigations, to detect the distribution and the related variance of the



hydraulic conductivity in three-dimensional domains. In particular, the optimized statistical parameters
(e.g., log conductivity variance and correlation range) of semivariograms are estimated using the modified
Gauss-Newton-Levenberg-Marquardt method. The Chaobai alluvial fan is used as a case study area.
Multiple data including downhole resistivity logging data, vertical electric soundings, well-bore
lithostratigraphies, TDS measurements, and depths to the water table are integrated to derive a dataset of
conductivity values in a three-dimensional setting. Log conductivity semivariograms fitted with
exponential functions are built-up for three facies, including fine sand, medium-coarse sand and gravel,
in each of the three zones into which the Chaobai fan is divided to guarantee local stationarity of the
statistical process. The composite semivariogram of the three facies has been derived for the two zones
where a sufficiently large number of samples are available. The $\log_{10}(K)$ configuration is simulated using
the sequential Gaussian simulation model based on statistic parameters of $\log_{10}(K)$ and the structure
suggested by a 3D hydrofacies simulation.

For the specific test case, the variance along the vertical direction of fine sand, medium-coarse sand, and
gravel decreases from the upper part of the alluvial fan, where the values amount to 0.23, 0.32, and 1.60,
to the lower portion of the Chaobai plan with values of 0.05, 0.126, and 0.62, respectively. This behavior
reflects the higher transport energy in the upper alluvial fan that causes a bad sediment sorting. In the
middle alluvial fan, the transport energy decreases and the sediments tend to be relatively good-sorted.
The variance of the gravel is larger than that of other lithologies. The different flow energy significantly
affected the coarse sediments in the vertical direction. Along the dip direction, the variance of three facies
(gravel, medium-coarse sand and fine sand) in the middle fan is larger than that in the lower fan. The



composite variance of $\log_{10}(K)$ in the vertical direction shows that the large heterogeneity in the upper
fan (with a value of 0.68) decreases in the lower zone.
The distribution of hydraulic conductivity is consistent with that of the facies. Hydraulic conductivity is
much larger in the upper zone than that in the lower part of the alluvial fan. This result provides valuable
insights for understanding the spatial variations of hydraulic conductivity and setting-up groundwater
flow, transport, and land subsidence models in alluvial fans.
Concluding, it is worth highlighting that we depicted an original method to detect the variance and
configuration of conductivity by fusing multiple-source data in three-dimensional domains. The proposed
approach can be easily used to statistically characterize the hydraulic conductivity of the various alluvial
fans that worldwide are strongly developed to provide high-quality water resources. We are aware of
some restrictions in the dataset available at the date for the Chaobai alluvial fan, for example the assumed
uniform distribution of TDS versus depth and the relatively small number of the conductivity samples in
the upper fan zone. Nonetheless, the proposed methodology will be re-applied in the near feature as soon
as new information will become available, thus allowing to improve the estimation accuracy of spatial
statistics parameters and the configuration of hydraulic conductivity in this Quaternary system so
important for the Beijing water supply.
**Data availability**
The geophysical measurements, borehole lithostratigraphies, and hydrogeological information in the
north part of Beijing Plain can be partly accessible by contacting Beijing Institute of Hydrogeology and
Engineering Geology.



**Author contribution**

Lin Zhu, Huili Gong and Zhenxue Dai derived the method of spatial variance and 3D configuration of conductivity, performed data analysis and wrote the draft manuscript. Gaoxuan Guo collected the geological and geophysical data, discussed the results. Pietro Teatini discussed the results, reviewed and revised the manuscript.

**Competing interests**

The authors declare that they have no conflict of interest.

**Acknowledgements**

This work was supported by the National Natural Science Foundation (No.41201420, 41130744) and Beijing Nova Program (No.Z111106054511097). Pietro Teatini was partially supported by the University of Padova, Italy, within the 2016 International Cooperation Program.

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



**Table 1 Values of the volumetric proportion for the various facies in three zones**

| Zone | Sub-clay and clay | Fine sand | Medium-coarse sand | Gravel |
|---|---|---|---|---|
| Zone 1 | 0.166 | 0.234 | 0.067 | 0.533 |
| Zone 2 | 0.409 | 0.286 | 0.065 | 0.240 |
| Zone3 | 0.503 | 0.328 | 0.106 | 0.063 |

**Table 2 Statistical data of logarithm hydraulic conductivity ($\log_{10}$(m/d)) in the three zones of the Chaobai alluvial fan**

| Zone | Parameter | Fine sand | Medium-coarse sand | Gravel |
|---|---|---|---|---|
| | **Mean** | **1.07** | **1.82** | **2.92** |
| Zone 1 | Minimum | -0.94 | 1.22 | 2.26 |
| | Maximum | 1.65 | 2.45 | 3.66 |
| | Proportion | 0.36 | 0.12 | 0.32 |
| | **Mean** | **0.42** | **1.17** | **2.65** |
| Zone 2 | Minimum | -2.22 | -0.23 | 0.95 |
| | Maximum | 1.22 | 2.07 | 3.38 |
| | Proportion | 0.23 | 0.14 | 0.31 |
| | **Mean** | **0.17** | **0.81** | **2.48** |
| Zone 3 | Minimum | -2.64 | -0.78 | 0.34 |
| | Maximum | 0.72 | 1.43 | 3.21 |
| | Proportion | 0.35 | 0.17 | 0.12 |





**Table 3 Optimized parameters in the fitting exponential function of $\log_{10}(K)$ semivariogram in vertical**
**direction for the various facies and zones**

| Zone | Parameter | Fine sand | | Medium-coarse sand | | Gravel | |
|---|---|---|---|---|---|---|---|
| | | Estimated value | Confidence Interval (95%) | Estimated value | Confidence Interval (95%) | Estimated value | Confidence Interval (95%) |
| Zone 1 | Variance | 0.23 | (0.19, 0.28) | 0.32 | ( 0.29, 0.34) | **1.60** | (1.41, 1. 81) |
| | Range (m) | 6.01 | (2.01, 20.52) | 8.01 | (1.53, 14.67) | 6.50 | (6.5, 12.84) |
| Zone 2 | Variance | 0.069 | (0.067, 0.070) | **0.14** | (0.13, 0.15) | **1.22** | (1.19, 1.24) |
| | Range (m) | 3.13 | (1.83, 4.42) | 8.27 | (3.61, 12.93) | 15.0 | (12.33, 17.67) |
| Zone3 | Variance | 0.05 | (0.047, 0.053) | **0.126** | (0.118, 0.135) | **0.62** | (0.54, 0.7) |
| | Range (m) | 6.52 | (2.19, 10.85) | 2.72 | (0.20, 6.55) | 5.98 | (0.20, 15.63) |




**Table 4 Variance of $\log_{10}(K)$ of different facies along the dip direction in Zone 2 and Zone 3**

| Zone | | Fine sand | Medium-coarse sand | Gravel |
|---|---|---|---|---|
| Zone 2 | Estimated value | **0.10** | **0.15** | **1.38** |
| | Confidence Interval (95%) | (0.059, 0.141) | (0.071, 0.228) | (1.14, 1.62) |
| Zone 3 | Estimated value | **0.045** | **0.068** | **0.48** |
| | Confidence Interval (95%) | (0.030, 0.0607) | (0.043, 0.093) | (0.22, 0.73) |








## Figure captions

Figure 1 Chaobai alluvial fan in the north of Beijing Plain. (a) Location of the study area and distribution of the field data. (b) Map of the hydraulic conductivity issued by Beijing Institute of Hydrogeology and Engineering Geology (2007). The location of the study area is shown in the inset.

Figure 2 Flowchart of the geostatistical methodology

Figure 3 Typical depth behaviour of resistivity and corresponding stratigraphy in the eastern part of Zone 2

Figure 4 Experimental (circle symbol) and model (solid line) semivariogram along the vertical direction for the various hydrofacies in the three zones. Notice that the range in the y-axis differs for sands and gravel lithologies in Zone 2 and Zone 3.

Figure 5 Experimental (circle symbol) and model (solid line) semivariogram along the dip direction for the various hydrofacies in Zone 2 and Zone 3.

Figure 6 Experimental (circle symbol) and model (solid line) composited semivariogram along the vertical direction for the three zones.

Figure 7 Distribution of hydrofacies (after Zhu et al., 2015a) and $\log_{10}(K)$ in the three-dimensional domain representing the Chaobai alluvial fan: (a) axonometric projection of the three-dimensional system and (b) vertical sections along the A-A', B-B', C-C' and D-D' alignments. The vertical exaggeration is 25. The selected cell size is 300 m in north-south and east-west directions and 5 m in vertical direction, with a total number of 747, 540 cells. The thickness of the simulated domain is 300 m.





**Figure 1**

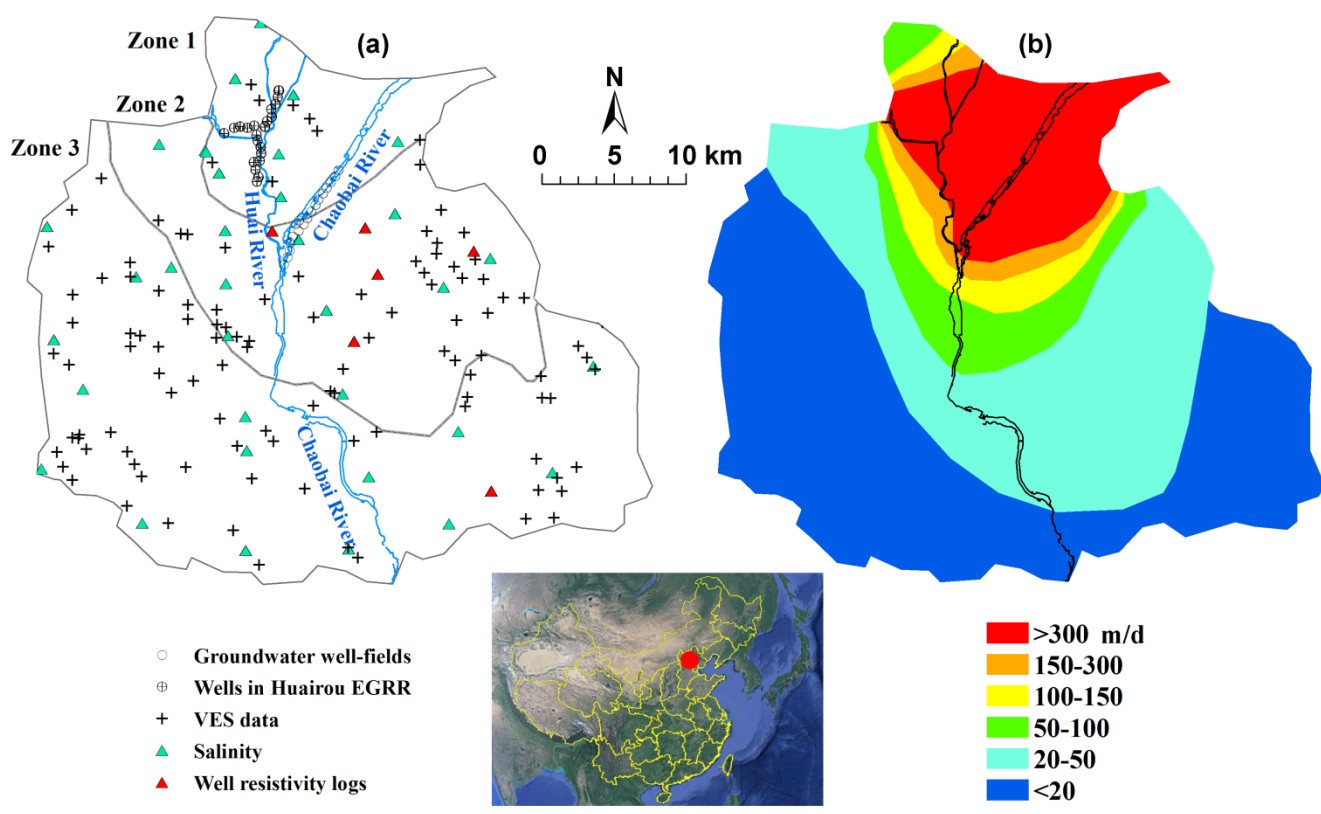






**Figure 2**

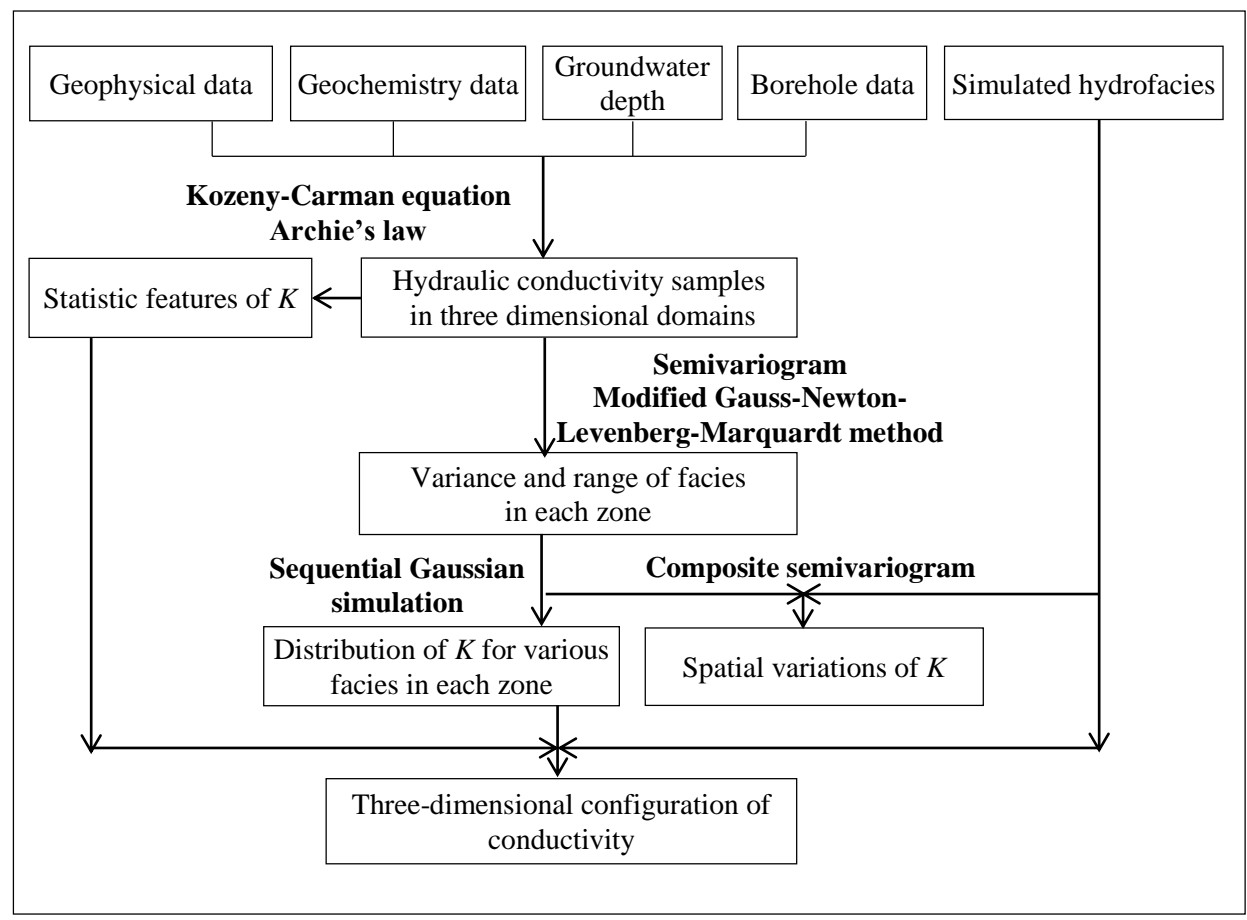







**Figure3**

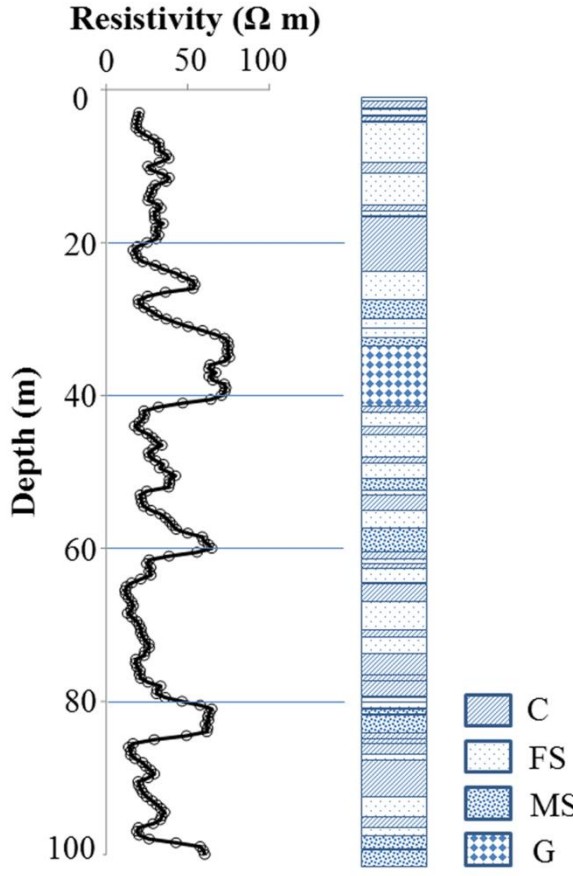











**Figure 4**

Zone1

Zone 2

Zone 3



**Figure 5**

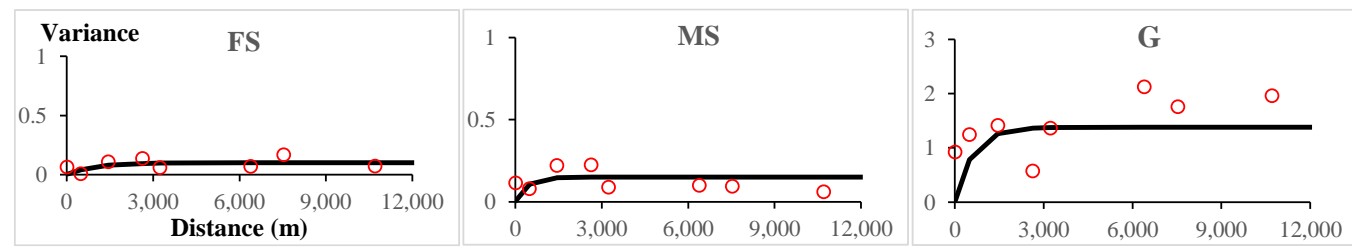

Zone 2

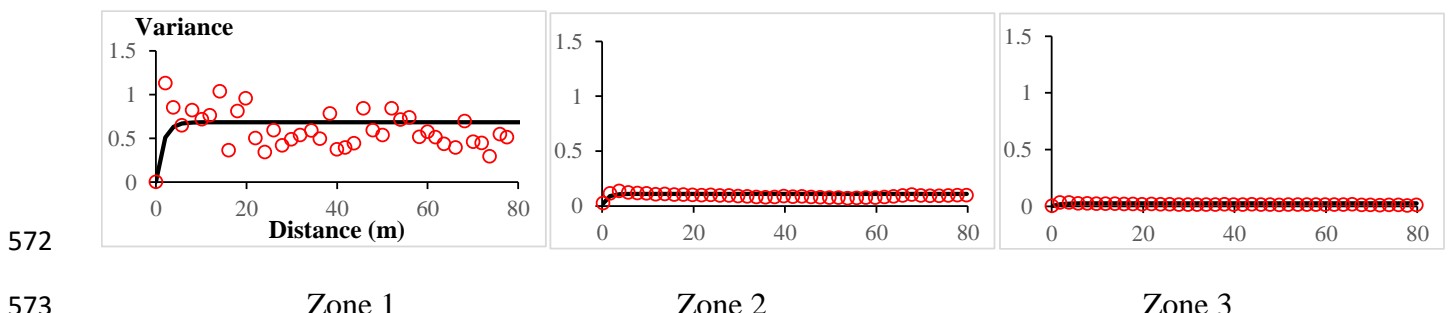

Zone 3

**Figure 6**

Zone 1                      Zone 2                      Zone 3





578    **Figure 7**

579

580