# Peer review of "Modelling 3D permeability distribution in alluvial fans using facies architecture and"

_Hydrology and Earth System Sciences, 2016_

## Referee Comment (RC1) · G. Weissmann (Referee) · 19 Sep 2016

**Review of Zhu et al. Modeling 3D permeability distribution in alluvial fans using facies architecture and geophysical acquisitions.**

In the full review and interactive discussion, the referees and other interested members of the scientific community are asked to take into account all of the following aspects:

1. Does the paper address relevant scientific questions within the scope of HESS? Yes. This manuscript presents use of multiple forms of data to address estimation of the distribution of hydraulic properties in an alluvial aquifer. Due to the highly heterogeneous nature of such aquifers, modeling these distributions is difficult.

2. Does the paper present novel concepts, ideas, tools, or data? In part, the combination of tools and geological and hydrogeological ideas makes this paper novel. Though others model heterogeneity in alluvial systems, these authors present a nice example of such work.

3. Are substantial conclusions reached? Nothing earth shaking, but still good results are presented, and others may be able to apply similar techniques to understanding heterogeneity in alluvial aquifers.

4. Are the scientific methods and assumptions valid and clearly outlined? Yes. This paper is well written and presented.

5. Are the results sufficient to support the interpretations and conclusions? Yes, generally. Some expansion on how lithofacies are simulated would be helpful.

6. Is the description of experiments and calculations sufficiently complete and precise to allow their reproduction by fellow scientists (traceability of results)? Yes. Though expansion on how lithofacies were defined would be helpful.

7. Do the authors give proper credit to related work and clearly indicate their own new/original contribution? Yes.

8. Does the title clearly reflect the contents of the paper? Yes, though you could specify that the geophysics is borehole geophysics. One may think that high-resolution seismic or other surface methods were used.

9. Does the abstract provide a concise and complete summary? Yes.

10. Is the overall presentation well structured and clear? Yes.

11. Is the language fluent and precise?  Yes.

12. Are mathematical formulae, symbols, abbreviations, and units correctly defined and used?  Yes

13. Should any parts of the paper (text, formulae, figures, tables) be clarified, reduced, combined, or eliminated?  Yes.  See comments below.

14. Are the number and quality of references appropriate?  Mostly yes.

15. Is the amount and quality of supplementary material appropriate?  None seen.

General comments:

The manuscript is well written and conceived.  The authors use an interesting mix of data sources (well logs, geophysical logs, and vertical electric soundings.  The manuscript offers interesting analyses and results, and with minor revisions would be a nice publication.  Below, I outline general comments and specific comments on the manuscript.

1. The authors call this an "alluvial fan", which in the current sedimentology/geomorphology literature this may not be strictly considered an alluvial fan.  Since it is longer than 30 km, this would be called a 'fluvial megafan' (e.g., Leier et al., 2005, Geology, v. 33, p. 289-292) or a "large distributive fluvial system (DFS)" (Hartley et al. 2010, Journal of Sedimentary Research, v. 80, p. 167-183).

2. In using the borehole geophysical tools (resistivity), the authors report resistivity values measured across the units.  Were these logs calibrated?  Are these values accurate given the calibration?  If they were not calibrated, the relative values between muds and sands would be important but the actual values are not significant.  If the logs were calibrated, some discussion of calibration procedures is important.  Also, these values are strongly influenced by the fluid conductivity.  Some discussion of fluid resistivity should be included.  The depth of water table is also important to note unless all data come from below the water table (resistivity properties in the vadose zone are different).  Line 161, where resistivity values are used in Archie's Law, makes it important that the logs were calibrated.  If the logs are not calibrated, Archie's Law is not appropriate to use.  Clarify this point.

3. The use of zones for modeling the system is appropriate and interesting.  However, progradation of fans often leads to these zones shifting position upward, where the coarser facies shown in Zone 1 overlie finer facies that you describe in Zone 2 (see Weissmann et al 2013, SEPM Special Publication 104, p. 131-147, for a discussion on this).  Do you see this pattern in the logs?  The abrupt boundary seen

between zones 1 and 2 in Figure 7 probably does not exist. This sharp transition could have been softened by using the results from the zone 1 simulation as conditioning for the zone 2 simulation. Likewise, the zone 2 simulation results could have been used as conditioning for the zone 3 simulation.

4. It isn't entirely clear how the lithofacies were placed into the variograms or modeled. Were lithofacies distributions modeled first, then variability added using variograms? Or, were K values assigned to the various lithofacies observed in well logs and lithologic logs, and then the variograms created from these K values? This must be clarified so others can apply your techniques. Looking at Figure 7, it appears as if you modeled lithofacies distributions first….how did you do this?

5. Additionally, your horizontal variograms don't show very much character. The fit is somewhat arbitrary. This is very common since the spacing between wells is often greater than the average widths/lengths of lithofacies, especially since strictly horizontal measures from logs may not indicate the actual paleo-horizontal. It's fine to show these models, but how sensitive were the results to these variograms (probably not very sensitive, but maybe).

6. Figure 7 indicates that the facies were modeled with dip direction in one orientation, thus facies near the apex area will not include a radiating pattern that emanates from the fan apex. This should be noted somewhere. Also, the authors could note that *a priori* orientation information from surface mapping of the fans could be used to create a model with channel orientations aimed at the apex (See work by Carle et al., where they are able to put varying orientations into the resulting models).

7. It is unclear how the vertical electric soundings were used, or what these data look like or what they show. This should be clarified.

Specific Comments:

1. Line 12: "heterogeneous" Heterogeneous in what? Hydraulic conductivity? Hydraulic properties?
2. Line 12, change "…which make difficult …" to "which make *it* difficult…"
3. Line 23: Change "bad" to "poor"
4. Line 35: Change "Conductivity distributions…" to "Hydraulic conductivity distributions…"
5. Line 36: Papers by Weissmann and Fogg (1999) and Weissmann et al (2002, 2004) also use conditional indicator geostatistical methods to model alluvial aquifers.
6. Line 90: Change "…exploration…" to "…exploitation…"
7. Line 100: remove the word "through"
8. Line 135: Change "…700 borehole lithostratigraphies were…" to "…700 borehole lithologic logs were…" My understanding is that these are the lithologic logs written by the drillers or well site geologists at time of drilling and are based on cuttings. Is that correct?
9. Line 141: "surrounding" Do you mean "in the area surrounding the sites of geophysical acquisitions."?
10. Line 156: What is the "representative" grain diameter"? D50? D10? How was this measured or estimated?

11. Line 192: "…vertical and dip directions…"  What was used in the strike direction (perpendicular to dip)?  Expected length scales would be expected to be less than what is used in the dip direction.
12. Line 248:  Change "…bad…" to "…poor…"  Grains are poorly sorted, not 'badly' sorted.
13. Line 254:  Change "…have a good sorting."  to "…are well sorted."
14. Line 260:  Change "…alike…" to "…similar…"
15. Lines 265-268:  This discussion of zone 2 does not make sense.  All of the zones have multiple flooding events depositing the sediments.  This medial zone does allow for greater preservation potential of finer sediments than the more proximal zone, where channel switching from the apex in the proximal zone tends to fill accommodation quickly and amalgamated channel belts are most likely preserved (see Weissmann et al. 2013, SEPM Special Publication 104, p. 131-147 for more details on structure of megafans and development of this structure through time).  A different explanation for the distributions of sediments and processes to develop these distributions is needed.
16. Line 293:  Change "…our average K values is gently…" to "…our average K values are gently…"
17. Line 293:  Change" …than these latter is likely…" to "…than these latter values are likely…"
18. Line 309:  "lithostratigraphies"  Do you mean "lithologic logs"?
19. Line 311:  "…are built-up…"  Do you mean "…were constructed…"?
20.  Line 320:  Change "..a bad sediment sorting"  to " …a poor sediment sorting"
21. Line 321:  Change "…relatively good sorted"  to "…relatively well sorted."
22.

---

## Referee Comment (RC2) · Anonymous Referee #2 · 4 Oct 2016

**Review of the paper titled: "Modelling 3D permeability distribution in alluvial fans using facies architecture and geophysical acquisitions" by Zhu et al.**

*Does the paper address relevant scientific questions within the scope of HESS?* Yes it does. Modelling physical heterogeneity in aquifers is still very important topic with myriad of theoretical and practical significances.

*Does the paper present novel concepts, ideas, tools, or data?* To some extent. The zonation approach to address the non stationarity of the K distribution is not new and it has been used in other previous studies. However, the integration of geological and geophysical data to derive K estimates is interesting. However, it is not completely novel since the same group of authors already presented the same approach in a recent previous publication (Zhu et al,. 2016b) based on the same dataset.

*Are substantial conclusions reached?* To some extent. Results indicate that the proposed method can be used to model *K* distribution in alluvial fans. However, this conclusion is mostly based on a qualitative assessment rather than a rigorous direct or indirect validation method. Moreover, it seems to me that a very similar conclusion was reached in their previous work (Zhu et al., 2016).

*Are the scientific methods and assumptions valid and clearly outlined?* To some extent. The methodology section could be expanded to better explain the criteria based on which the four facies were chosen, and how the lithological model in Figure 7A was developed.

*Are the results sufficient to support the interpretations and conclusions?* Yes.

*Is the description of experiments and calculations sufficiently complete and precise to allow their reproduction by fellow scientists (traceability of results)?* To some extent. A description of how lithofacies were defined and simulated could be beneficial.

*Do the authors give proper credit to related work and clearly indicate their own new/original contribution?* Generally yes. However, they should explain the novelty of this work with respect to their previous one.

*Does the title clearly reflect the contents of the paper?* Yes.

*Does the abstract provide a concise and complete summary?* Yes.

*Is the overall presentation well structured and clear?* To some extent. See comments.

*Is the language fluent and precise?* Some descriptions sound colloquial.

*Are mathematical formulae, symbols, abbreviations, and units correctly defined and used?* Yes.

*Should any parts of the paper (text, formulae, figures, tables) be clarified, reduced, combined, or eliminated?* Yes. See comments.

*Are the number and quality of references appropriate?* In part. See comments.

*Is the amount and quality of supplementary material appropriate?* None provided.

General comments

Although the methods described here are not new, the paper presents an interesting case study that could be worth of publication. However, I think that major revisions are necessary to address the following issues:

1) What is the novelty of the results presented in this work with respect to the recent publication by the same group of authors (i.e., Zhu et al., 2016b)? This not clear to me, and I think it should appear in the introduction of this paper, together with the motivation for expansion or improvement (if any).

2) I do not understand how the facies C, FS, MS and G have been defined. Are the names of these units referring to the prevalent grain size value? The reason for this question is that the K values estimated for these units in the three zones show inconsistencies. In table 2, for instance, how do the authors explain that the facies called "Medium-coarse sand" in zone 3 is less conductive (0.81 vs. 1.07) than the facies called "Fine sand" in zone 1? Is the "fine sand" unit in zone 1 the same as the "fine sand" unit in zone 3? If yes, then why the mean K value is about 8 times larger? What I mean is that a deposit consisting largely of fine sand is a "fine sand" regardless of its location with respect to the apex of the alluvial fan. The same applies to all the other units. It is the proportions of these facies that changes with distance from the apex, and these changes are responsible for the non stationarity of the K distribution. The average K value of the lithofacies (e.g., "fine sand") should be consistent (plus minus uncertainty) between zones. Please clarify this point because it seems to me a major flaw of the proposed methodology. I also wonder if a different classification with more units would be also more appropriate in this case. In particular, fan deposits especially in the proximal part (your zone 1) are also characterized by debris flow deposits (matrix supported gravel). This type of deposits has been ignored.

3) I think it would vastly improve the impact of this analysis if the authors can include some quantitative assessment of the accuracy of the reconstructed K distribution. For instance, it seems that there is enough data to apply a split sample validation test. A comparison with results in which K is assumed stationary would also be beneficial.

Specific comments

1) Line 34. The reference Zappa et al. (2006) refers to a different depositional environment. Please use a more specific reference.

2) Line 35. I suggest to add "hydraulic".

3) Line 65. Insert a period before "The Chaobai".

4) Line 79. Deposited instead of "laid down"

5) Line 156. Please specify what you consider as representative grain size diameter. Is it the $d_{10}$?

6) Lines 161 – 162. Only ranges are provided for the parameters. What specific values have been used? Why? Please explain.

7) Equations 2 and 3. How do the systematic errors and uncertainty in the parameters associated with these empirical equations affect the uncertainty in the K estimations? Some comment on this would be beneficial.

8) Line 169. I suggest to show the histograms of the *K* values within each facies to justify the lognormality assumption.

9) Line 172. What information? Grain size analyses?

10) Line 180. Are those data different then the data used in this work? The fact that you say that you found consistency suggests that they are different, but then I wonder why these 694 boreholes were not considered. Please clarify.

11) Line 217. I understood that the volumetric proportions $p_k$ were derived from the borehole data rather than estimated through inversion. Please clarify.

12) Line 244. Variance of what?

13) Line 246. "… is highly uncertain"

14) Line 248 - 249. This sentence does not find correspondence in your analysis. On the other hand it confirms my doubt that the facies classification at the basis of the proposed methodology is not correct (see general comment 2). You rightly write "The deposits consist of wide ranges of sediment categories and grain sizes" to justify the fact that *K* has higher variance. But this is not the case in this work because the presented analysis is based on the assumption of only four (even three here) units G, FS, MS. So it seems to me that in order to include that variability you are talking about, your units do not actually represent a specific lithology as the names imply but a wider range of lithologies. For instance, your unit called "Fine sand" may actually include deposits that would be classified as fine sands as well as fine gravels. Am I wrong?

15) Line 250. Heterogeneity of what?

16) Line 254. Sorting and grain size are not the same. A poorly sorted sediment can still have a very high *K*.

17) Line 263. I suggest to provide the variance also for zone 1. After all, if I understood correctly, there are 102 samples. This is not such a small number.

18) Lines 266 – 269. Please revise the sentence. The meaning is not clear.

19) Lines 274 - 276. Same comment as #14.

20) Lines 280 – 281. Is this conclusion not obvious? I understood that the SGS realizations of $K$ are mapped on the basis of the facies model (Figure 7a).

21) Line 281 – 282. This should not be caused by assigning larger average $K$ of three units. This should be the consequence of the fact that coarser units are more frequent in this zone and therefore the average $K$ is larger. The $K$ distribution is the product of the lithological heterogeneity; it is not the opposite as it is implied here and in general in the paper.

22) Lines 293 – 296. It depends. Are you considering arithmetic or geometric average?

---

## Author Comment (AC1) · 8 Oct 2016

Comment 1: The authors call this an "alluvial fan", which in the current sedimentology/geomorphology literature this may not be strictly considered an alluvial fan. Since it is longer than 30 km, this would be called a 'fluvial megafan' (e.g., Leier et al., 2005, Geology, v. 33, p. 289-292) or a "large distributive fluvial system (DFS)" (Hartley et al. 2010, Journal of Sedimentary Research, v. 80, p.167-183).

Response: We agree with this comment. The huge alluvial fan we dealt with is really a "large distributive fluvial system" or "megafan". In our revised text, we changed the term "alluvial fan" to "alluvia megafan" when describing our site model; when discussing about general alluvial fans, we still used the general term. We also cited these two papers the reviewer provided as: 1. Leier, A.L., P. G. DeCelles, J. D. Pelletier, Mountains, monsoons, and megafans, Geology, v. 33, p. 289-292, 2005. 2. Hartley, A.J., Weissmann, G.S., Nichols, G.J., and Warwick, G.L., Distributive fluvial systems: characteristics, distribution, and controls on development: Journal of Sedimentary Research, v. 79, p. 167-183, 2010.

Comment 2: In using the borehole geophysical tools (resistivity), the authors report resistivity values measured across the units. Were these logs calibrated? Are these values accurate given the calibration? If they were not calibrated, the relative values between muds and sands would be important but the actual values are not significant. If the logs were calibrated, some discussion of calibration procedures is important. Also, these values are strongly influenced by the fluid conductivity. Some discussion of fluid resistivity should be included. The depth of water table is also important to note unless all data come from below the water table (resistivity properties in the vadose zone are different). Line 161, where resistivity values are used in Archie's Law, makes it important that the logs were calibrated. If the logs are not calibrated, Archie's Law is not appropriate to use. Clarify this point.

Response: In this study, we assumed that clay fraction is negligible in the faces of gravel (G), medium-coarse sand (MS), and fine sand (FS) (new Line179-180). To calibrate the resistivity loggings data, we used the resistivity located in the middle of the facies block, where the resistivity is approximately the real resistivity. For VES data, we compared the inversed resistivity with the observed stratigraphic information (see Fig. 3 and new Fig. 4). Calibration is obtained by comparing the VES outcome with direct investigation, e,g. well stratigraphies and the inversed resistivity can reflect the difference of facies: the thick gravel layer has larger resistivity while the fine sand and clay layers have relatively smaller resistivity. The fluid conductivity was estimated by using total dissolved solids (TDS) and temperature data. Because of the relatively limited dataset and the observed small variability, in this paper the TDS variations in the vertical direction were neglected (Line147-148). That is, there is one TDS value at one logging position (vertically). We focused on the resistivity data below water table. The inverted resistivity from the VES data and the logging resistivity data were used in Archies' Law.

Comment 3: The use of zones for modeling the system is appropriate and interesting. However, progradation of fans often leads to these zones shifting position upward, where the coarser facies shown in Zone 1 overlie finer facies that you describe in Zone 2 (see Weissmann et al 2013, SEPM Special Publication 104, p. 131-147, for a discussion on this). Do you see this pattern in the logs? The abrupt boundary seen between zones 1 and 2 in Figure 7 probably does not exist. This sharp transition could have been softened by using the results from the zone 1 simulation as conditioning for the zone 2 simulation. Likewise, the zone 2 simulation results could have been used as conditioning for the zone 3 simulation.

Response: Yes, from the representative borehole data (5 logs in Zone 1) we see that the coarser facies in Zone 1 overlie finer facies (which has a small volumetric proportion and thickness). To smooth the abrupt boundary between Zone 1 and Zone 2, we used conditional facies data from boreholes (5 logs in Zone 1 and 15 logs in Zone 2) near the boundary to make the facies change gradually. But, since the volumetric proportions for sub-clay, clay and gravel change a lot from Zone 1 to Zone 2, we can still see that boundary. Next step we will follow this reviewer's suggestions to improve the simulated facies distributions near the zone boundaries by using some simulation results of Zone 1 at the boundary as conditional data for the simulation of Zone 2. Two references about indicator simulations in highly heterogeneous formations are cited in this section as: 1. Weissmann, G.S., Hartley, A.J., Scuderi, L.A., Nichols, G.J., Davidson, S.K., Owen, A., Atchley, S.C., Bhattacharyya, P., Chakraborty, T., Ghosh, P., Nordt, L.C., Michel, L., and Tabor, N.J. Prograding distributive fluvial systems–geomorphic models and ancient examples, in Driese, SG, and Nordt, LC (eds), New Frontiers in Paleopedology and Terrestrial Paleoclimatology, SEPM Special Publication No. 104, p. 131-147, 2013. 2. Maghrebi, M., Jankovic, I., Weissmann, G.S., Matott, L.S., Allen-
King, R.M., and Rabideau, A.J., Contaminant tailing in highly heterogeneous porous formations: Sensitivity on model selection and material properties. J. of Hydrol., 531, 149-160, 2015.

Comment 4: It isn't entirely clear how the lithofacies were placed into the variograms or modeled. Were lithofacies distributions modeled first, then variability added using variograms? Or, were K values assigned to the various lithofacies observed in well logs and lithologic logs, and then the variograms created from these K values? This must be clarified so others can apply your techniques. Looking at Figure 7, it appears as if you modeled lithofacies distributions first. . ..how did you do this?

Response: Yes, the lithofacies distribution was modeled first. Then, the distributions of $\log10(K)$ for three facies including G, MS, and FS were simulated, respectively, using SGSIM with the parameter determined from individual semivariograms. The 3D cells of the distributions of $\log10(K)$ is the same as that of the lithofacies indicator model. And then, since each cell is characterized by specific facies indicator and zone indices, its conductivity was assigned or mapped based on the corresponding facies, zone numbers and the 3D SGSIM simulated conductivity values. Finally, since sub-clay and clay are generally characterized by a low hydraulic conductivity value, a uniform conductivity value equal to 0.0001 m/d was set to all the C cells (new Line 244-247). The procedure flowchart is already provided in Figure 2.

Comment 5: Additionally, your horizontal variograms don't show very much character. The fit is somewhat arbitrary. This is very common since the spacing between wells is often greater than the average widths/lengths of lithofacies, especially since strictly horizontal measures from logs may not indicate the actual paleo horizontal. It's fine to show these models, but how sensitive were the results to these variograms (probably not very sensitive, but maybe).

Response: Yes, the horizontal variograms are fitted with some degree of arbitrary due to the sparse horizontal samplings or larger well distances. By using the variances estimated from the inversions of vertical variograms as prior information, the estimated variances and ranges of log10(K) in horizontal direction are reasonably constrained. We added a few sentences to discuss the uncertainty of the estimated horizontal ranges.

Comment 6: Figure 7 indicates that the facies were modeled with dip direction in one orientation, thus facies near the apex area will not include a radiating pattern that emanates from the fan apex. This should be noted somewhere. Also, the authors could note that a priori orientation information from surface mapping of the fans could be used to create a model with channel orientations aimed at the apex (See work by Carle et al., where they are able to put varying orientations into the resulting models). Response: In the description of Figure 7 (new Figure 8), we added a sentence to indicate that "since we simulate the dip direction along one orientation (along the main water flow direction), the simulated facies in the fan apex do not show a radiating pattern. More information about simulating the radiating pattern can be found from Carle and Fogg (1997) and Fogg et al. (1998)".

Fogg, G.E., C. D. Noyes, S. F. Carle, Geologically based model of heterogeneous hydraulic conductivity in an alluvial setting, Hydrogeol. J. , 6(1), 131-143, 1998.

Comment 7: It is unclear how the vertical electric soundings were used, or what these data look like or what they show. This should be clarified.

Response: We added Figure 4 to show the inverted resistivity from VES compared with the lithologic data.

Specific Comments:

1. Line 12: "heterogeneous" Heterogeneous in what? Hydraulic conductivity? Hydraulic properties?

Response: Changed. Here means heterogeneous in hydraulic properties.

2. Line 12, change ". . .which make difficult . . ." to "which make it difficult. . ."

Response: Suggestion followed.

3. Line 23: Change "bad" to "poor"

Response: Changed.

4. Line 35: Change "Conductivity distributions..." to "Hydraulic conductivity distributions..."

Response: Done.

5. Line 36: Papers by Weissmann and Fogg (1999) and Weissmann et al (2002, 2004) also use conditional indicator geostatistical methods to model alluvial aquifers.

Response: The suggested references have been added in the Introduction section.

6. Line 90: Change "...exploration..." to "...exploitation..."

Response: Changed.

7. Line 100: remove the word "through"

Response: Changed.

8. Line 135: Change "...700 borehole lithostratigraphies were..." to "...700 borehole lithologic logs were..." My understanding is that these are the lithologic logs written by the drillers or well site geologists at time of drilling and are based on cuttings. Is that correct?

Response: Yes, you are right. The sentence was revised.

9. Line 141: "surrounding" Do you mean "in the area surrounding the sites of geophysical acquisitions."?

Response: Yes, the sentence was revised.

10. Line 156: What is the "representative" grain diameter"? D50? D10? How was this measured or estimated?

Response: The d(x,y,z) is the median grain diameter (D50) of the facies, which is determined on basis of the measurements of the lithologic samples. The sentence was changed.

11. Line 192: "...vertical and dip directions..." What was used in the strike direction (perpendicular to dip)? Expected length scales would be expected to be less than what is used in the dip direction.

Response: The semivariogram in strike direction is not calculated in this study. When simulating the distributions of log10(K), there is an assumption that the conductivity in dip/strike direction is the same with that in horizontal direction.

12. Line 248: Change "...bad..." to "...poor..." Grains are poorly sorted, not 'badly' sorted.

Response: Changed.

13. Line 254: Change "...have a good sorting." to "...are well sorted."

Response: Changed.

14. Line 260: Change "...alike..." to "...similar..."

Response: Changed.

15. Lines 265-268: This discussion of zone 2 does not make sense. All of the zones have multiple flooding events depositing the sediments. This medial zone does allow for greater preservation potential of finer sediments than the more proximal zone, where channel switching from the apex in the proximal zone tends to fill accommodation quickly and amalgamated channel belts are most likely preserved (see Weissmann et al. 2013, SEPM Special Publication 104, p. 131‐147 for more details on structure of megafans and development of this structure through time). A different explanation for the distributions of sediments and processes to develop these distributions is needed.

Response: We revised the discussions in former Lines 265-268 by taking account the complex structures of the megafan. We indicate that "Zone 2 extends from the fan apex area with much larger area which allows for greater preservation potential of finer sediments (such as medium-coarse sand (MS), fine sand (FS), and clay or sub clay (C)) than the more proximal Zone 1. Therefore, in Zone 2 the volumetric proportions for these three facies increase while that of gravel decreases. The estimated ranges of G and MS are increased, respectivelyïijĹrange of FS decreased. In Zone 3, the range difference among the three facies decreases gradually. The estimated range of FS is about 6.0 m, which is twice as much as that of MF. The spatial variation of the structure parameters of three facies causes the large changes of the correlation ranges from Zone 1 to Zone 3.

16. Line 293: Change ". . .our average K values is gently. . ." to ". . .our average K values are gently. . ."

Response: Changed.

17. Line 293: Change" . . .than these latter is likely. . ." to ". . .than these latter values are likely. . ."

Response: Changed.

18. Line 309: "lithostratigraphies" Do you mean "lithologic logs"?

Response: Changed.

19. Line 311: ". . .are built‐up. . ." Do you mean ". . .were constructed. . ."?

Response: Changed.

20. Line 320: Change "..a bad sediment sorting" to " . . .a poor sediment sorting"

Response: Changed.

21. Line 321: Change ". . .relatively good sorted" to ". . .relatively well sorted."

Response: Done.

Please also note the supplement to this comment:
http://www.hydrol-earth-syst-sci-discuss.net/hess-2016-373/hess-2016-373-AC1-supplement.pdf

———————————————————

[Figure]

Zone 1

Zone 2

Zone 3

**(a)**

N

5   10 km

*Chaobai River*

*Huai River*

*Chaobai River*

**(b)**

○   Groundwater well-fields
⊕   Wells in Huairou EGRR
+   VES data
▲   Salinity
▲   Well resistivity logs

■ >300  m/d
■ 150-300
■ 100-150
■ 50-100
■ 20-50
■ <20

**Fig. 1.** Chaobai alluvial fan in the north of Beijing Plain.

Figure 2

[Figure]

**Fig. 2.** Flowchart of the geostatistical methodology

[Figure]

**Fig. 3.** Typical depth behaviors of resistivity and corresponding stratigraphy in the eastern part of Zone 2

[Figure]

**Fig. 4.** Inversed resistivity and corresponding stratigraphy in Zone 1

![Figure showing semivariograms for three zones, each with FS, MS, and G panels plotting Variance vs Distance (m)]

[revised manuscript text omitted]

Zone1

Zone 2

Zone 3

**Figure 6**

[Figure]

                                   Zone 2

[Figure]

                                   Zone 3

**Figure 7**

[Figure]

    Zone 1                      Zone 2                      Zone 3

**Figure 8**

[Figure]

| | |
|---|---|
| ■ | gravel |
| ■ | medium-coarse sand |
| ■ | fine sand |
| ■ | sub-clay and clay |

| | |
|---|---|
| 3.75 | |
| 1.81 | |
| -0.12 | |
| -2.06 | |
| -4.00 | $\log_{10}(K)$ |

---

## Referee Comment (RC3) · Anonymous Referee #3 · 11 Oct 2016

1) Scientific Significance Does the manuscript represent a substantial contribution to scientific progress within the scope of this journal (substantial new concepts, ideas, methods, or data)?

Poor

Concepts, ideas, and methods are not new. The claim of an "original method" by the authors is unfounded. Every method used has been previously published and implemented: Dividing a domain into zones to do geostatistical modeling is not original; use of geophysical data to derive facies or hydraulic parameters is not original ; assumptions of "K distributions are local stationary" and computing the log10(K) semivariogram are decades-old concepts. The paper generally reads like documentation of a work assignment, not scientific progress.

2) Scientific Quality Are the scientific approach and applied methods valid? Are the results discussed in an appropriate and balanced way (consideration of related work, including appropriate references)?

Poor

In judging scientific quality, consider the scientific method: systematic observation, measurement, and experiment, and the formulation, testing, and modification of hypotheses.

Granted, the paper does implement some systematic observation and measurement and proceeds to set up an "experiment" of sorts by producing geostatistical realizations of hydraulic conductivity. However, it is not clear at all what the hypotheses of the paper are and what the "experiment" will actually be: A flow model or a transport model for what use? A calibration/validation exercise to what observations? The paper simply lacks scientific completeness in formulation and testing of hypotheses.

The authors seem to be advocating that an "original method" constitutes science or, perhaps, a hypothesis of being "original" is science, but even that is not necessarily true especially since the authors' claim of being "original" is debatable. The closest statement to a hypothesis I could find is given at the end of section 2.1:

"The characterization of the distribution and spatial variability of hydraulic conductivity is vital for an optimal use of the limited water resources in this area."

This statement isn't new or "original" except perhaps at the particular area of study in China. More importantly, this hypothesis is not tested in the paper! Instead, the paper is consumed with mundane documentation of its observations and methods and preparation of an experiment that is never executed. The paper could have tested whether its methods are actually "vital for an optimal use of the limited water resources in this area" (e.g. by flow or transport modeling with comparison to water level or chemistry

data, i.e. observations). A scientific result would be proof that the author's methods are better than a typical effective-K model for determination of some "vital" information about the aquifer system. Perhaps the authors plan to do this in another paper, but that does not matter. The existing paper just does not constitute good science on its own.

From a hydrogeologic perspective, important qualities are lacking in the representation of alluvial fan heterogeneity: (1) there is no directional non-stationarity (e.g. no radial variability of the depositional major axis; no stratigraphic dip), (2) there are abrupt, unrealistic discontinuities between zone (e.g. facies occurrences abruptly terminate and the edge of a zone, like a fault), and (3) the zonal approach leads to unrealistic transitions in geometrical properties (e.g. thickness of gravel deposits). For all the claims of being an "original method" by combining different methods, the paper does not seem to pay close attention to methods of geology.

3) Presentation Quality Are the scientific results and conclusions presented in a clear, concise, and well structured way (number and quality of figures/tables, appropriate use of English language)?

Poor

Again, the paper is really lacking in actual scientific results (i.e. results of hypothesis testing). The paper is full of documentation of what was done to analyze data and make geostatistical realizations, including re-hashing of old methods with obvious weighting to referencing of the authors' previous publications. Even if the conclusion that "it is worth highlighting we depicted an original method..." were true, this does not constitute good science on its own. The claim of "Fusing multiple-source data" isn't necessarily science on its own since it is routine practice in the earth sciences.

Figures 5 & 6 were never referenced in the text. Figures 4-6 are difficult to interpret without labeling of y-axis units and use of variable scales in Figure 4 & 5. Discussion of dip and strike direction model parameters other than variance is lacking. Figure 7 has no scale. These are key elements to geostatistical modeling, yet this information

[Figure]

was poorly presented.

In terms of documenting what the authors did, the paper is a reasonable piece of communication of the caliber of an institutional report (which would need further revision in regard to Figure 4-7 as noted above and use of English language).

For final publication, the manuscript should be

Rejected

---

## Author Comment (AC2) · 25 Oct 2016

Thanks for the constructive comments from Reviewer 2. We carefully revised the texts by incorporating his/her comments one by one. The detailed revision is presented in the response to each comment. Comment 1) What is the novelty of the results presented in this work with respect to the recent publication by the same group of authors (i.e., Zhu et al., 2016b)? This not clear to me, and I think it should appear in the introduction of this paper, together with the motivation for expansion or improvement (if any). Response: The simulation of 3D hydraulic conductivity in Zhu et al. (2016b) was obtained by combing the interpolated resistivity and the stochastic simulated facies through empirical equation (New line 63-66). Moreover, in the previous work only

VES data was used to get the porosity and the hydraulic conductivity was converted from the porosity data by using an empirical equation. In this paper, we constructed the 3D hydraulic conductivity by coupling the indicator facies simulations and sequential Gaussian conductivity simulations within each facies using the spatial geostatistical parameters deduced from the log conductivity semivariograms of different facies in different zones. The geophysical data are interpreted for computing the hydraulic conductivity distributions at different sampling locations. The novelty of this work is to develop an integrated approach to reconstruct the three-dimensional configuration of conductivity in the alluvial fan by coupling the hydrofacies indicator simulations with conductivity spatial heterogeneity simulations by using the hydrogeological and geophysical measurements or resistivity loggings and electrical soundings. The newly collected geophysical data combining with the sequential Gaussian simulations reduce greatly the uncertainty of the reconstructed three-dimensional conductivity fields. Finally, also the scale of application is very different, with the previous work focused on a small test area and this one to the whole megafan of the Chaobai river. Comment 2) I do not understand how the facies C, FS, MS and G have been defined. Are the names of these units referring to the prevalent grain size value? The reason for this question is that the K values estimated for these units in the three zones show inconsistencies. In table 2, for instance, how do the authors explain that the facies called "Medium-coarse sand" in zone 3 is less conductive (0.81 vs. 1.07) than the facies called "Fine sand" in zone 1? Is the "fine sand" unit in zone 1 the same as the "fine sand" unit in zone 3? If yes, then why the mean K value is about 8 times larger? What I mean is that a deposit consisting largely of fine sand is a "fine sand" regardless of its location with respect to the apex of the alluvial fan. The same applies to all the other units. It is the proportions of these facies that changes with distance from the apex, and these changes are responsible for the non-stationarity of the K distribution. The average K value of the lithofacies (e.g., "fine sand") should be consistent (plus minus uncertainty) between zones. Please clarify this point because it seems to me a major flaw of the proposed methodology. I also wonder if a different classification with more units would

be also more appropriate in this case. In particular, fan deposits especially in the proximal part (your zone 1) are also characterized by debris flow deposits (matrix supported gravel). This type of deposits has been ignored. Response: The hydrofacies (e.g., C, FS, MS and G) are defined qualitatively based on the sedimentary structures, borehole lithological descriptions, and grain sizes, while the conductivity samples are then deduced from geophysical measurements for each facies at each zone. Since the clay contents from zone 1 to zone 3 are increased due to the changes in the sediment transport conditions, for the same facies we also found this trend and the overall hydraulic conductivities are decreased from zone 1 to zone 3. As this reviewer stated, "these changes are responsible for the non-stationarity of the K distribution". Therefore, we used multi-zone approach to overcome the overall non-stationarity and to assume a local stationary for each zone. Although we still call the similar faces at different zones with the same name, but the structure and statistical parameters for each facies at different zones are quite different. We simulated the facies and the hydraulic conductivity sequentially with the estimated structure and statistical parameters from zone 1 to zone 3 and the final constructed three-dimensional conductivity field can represent the sedimentary and hydrogeological conditions in this alluvial fan. We added a few sentences to explain the statistical parameters listed in Table 2. Comment 3) I think it would vastly improve the impact of this analysis if the authors can include some quantitative assessment of the accuracy of the reconstructed K distribution. For instance, it seems that there is enough data to apply a split sample validation test. A comparison with results in which K is assumed stationary would also be beneficial. Response: The distributions of hydraulic conductivity are obtained through coupling facies indicator simulations and conductivity sequential Gaussian simulations, which represent the spatial features of the sediments including facies lengths and volumetric proportions and the hydraulic conductivity heterogeneity including log10K mean, variance and correlation scales. The simulated three-dimensional conductivity field was validated with the values provided by the Beijing Institute of Hydrogeology and Engineering Geology (2007) based on a number of pumping (Line 318-325). Specific comments 1) Line 34. The reference Zappa et al. (2006) refers to a different depositional environment. Please use a more specific reference. Response: Suggestion followed. We added a specific reference here as: Weissmann, G.S., S.F. Carle, G.E. Fogg, Three-dimensional hydrofacies modeling based on soil surveys and transition probability geostatistics, Water Resour. Res., 35(6), 1761–1770, 1999. 2) Line 35. I suggest to add "hydraulic". Response: Changed. 3) Line 65. Insert a period before "The Chaobai". Response: Added. 4) Line 79. Deposited instead of "laid down" Response: Changed. 5) Line 156. Please specify what you consider as representative grain size diameter. Is it the d10? Response: Changed. d(x,y,z) is the median grain diameter (D50, mm). 6) Lines 161 – 162. Only ranges are provided for the parameters. What specific values have been used? Why? Please explain. Response: ïĄą is equal to 1. In the upper part of the alluvial fan (Zone 1 and Zone) m is set equal to 1.3 because the sand is unconsolidated. In Zone 3. m is set to 1.7 which reflects slightly compressed or cemented sandstones (Niwas et al. 2011). 7) Equations 2 and 3. How do the systematic errors and uncertainty in the parameters associated with these empirical equations affect the uncertainty in the K estimations? Some comment on this would be beneficial. Response: We added a sentence to discuss the parameter uncertainty estimated from this empirical equations as: Note that the parameters associated with equations (2) and (3) are site specific and the application these equations to other sites will need a re-adjustment of the related parameters. 8) Line 169. I suggest to show the histograms of the K values within each facies to justify the lognormality assumption. Response: The histograms of log10K within each facies are given in Figure 5 in our new version paper.

9) Line 172. What information? Grain size analyses? Response: The information is the lithological descriptions and grain size distributions collected from different boreholes. 10) Line 180. Are those data different then the data used in this work? The fact that you say that you found consistency suggests that they are different, but then I wonder why these 694 boreholes were not considered. Please clarify. Response: There are 113 borehole data used in this work which are chosen from the original 694 boreholes. The lithological information in a buffer zone of 200 m

around the VES locations has been used to represent the actual facies distribution in the area surrounding the sites of the geophysical acquisitions (New line 154-155, original line 140-141). 11) Line 217. I understood that the volumetric proportions pk were derived from the borehole data rather than estimated through inversion. Please clarify. Response: The facies transition probability models are calculated on basis of the borehole lithological description data. The analytical equation (Equation 9) of transition probability was used to fit the sample transition probability and to estimate the volumetric proportions and indicator correlation lengths. 12) Line 244. Variance of what? Response: Changed. Variance of the log conductivity. 13) Line 246. ". . . is highly uncertain" Response: Changed. 14) Line 248 - 249. This sentence does not find correspondence in your analysis. On the other hand it confirms my doubt that the facies classification at the basis of the proposed methodology is not correct (see general comment 2). You rightly write "The deposits consist of wide ranges of sediment categories and grain sizes" to justify the fact that K has higher variance. But this is not the case in this work because the presented analysis is based on the assumption of only four (even three here) units G, FS, MS. So it seems to me that in order to include that variability you are talking about, your units do not actually represent a specific lithology as the names imply but a wider range of lithologies. For instance, your unit called "Fine sand" may actually include deposits that would be classified as fine sands as well as fine gravels. Am I wrong? Response: As we responded to the general comment 2, the hydrofacies (e.g., C, FS, MS and G) are defined qualitatively based on the sedimentary structures, borehole lithological descriptions, and grain sizes, while the conductivity samples are then deduced from geophysical measurements for each facies at each zone. Since the clay contents from zone 1 to zone 3 are increased due to the changes in the sediment transport conditions, for the same facies we also found this trend and the overall hydraulic conductivities are decreased from zone 1 to zone 3. As this reviewer stated, "these changes are responsible for the non-stationarity of the K distribution". Therefore, we used multi-zone approach to overcome the overall non-stationarity and to assume a local stationary for each

zone. Although we still call the similar faces at different zones with the same name, the structure and statistical parameters for each facies at different zones are quite different. We simulated the facies and the hydraulic conductivity sequentially with the estimated structure and statistical parameters from zone 1 to zone 3 and the final constructed three-dimensional conductivity field can represent the sedimentary and hydrogeological conditions in this alluvial fan. 15) Line 250. Heterogeneity of what? Response: Changed. Heterogeneity of hydraulic conductivity. 16) Line 254. Sorting and grain size are not the same. A poorly sorted sediment can still have a very high K. Response: Yes, we agree. We changed the related terms. 17) Line 263. I suggest to provide the variance also for zone 1. After all, if I understood correctly, there are 102 samples. This is not such a small number. Response: Yes, 102 samples are not a small number. These samples are deduced from eight positions on horizontal plane (Figure 1). When we calculated semivariograms in dip direction using these samples, there are too many zero values and then we ignored the semivariograms and the variances in dip direction in Zone 1. As this reviewer suggested, our next-step study is to collect more hydraulic conductivity to provide the variance in zone 1. 18) Lines 266 – 269. Please revise the sentence. The meaning is not clear. Response: Revised (New line 293-296). 19) Lines 274 - 276. Same comment as #14. Response: We revised the sentences here to incorporate this comment. 20) Lines 280 – 281. Is this conclusion not obvious? I understood that the SGS realizations of K are mapped on the basis of the facies model (Figure 7a). Response: The three-dimensional hydraulic conductivity was obtained by combining the facies model and the sequential Gaussian simulations of hydraulic conductivity for each facies. Detail information on re-constructing the K is given in new line 252-262. 21) Line 281 – 282. This should not be caused by assigning larger average K of three units. This should be the consequence of the fact that coarser units are more frequent in this zone and therefore the average K is larger. The K distribution is the product of the lithological heterogeneity; it is not the opposite as it is implied here and in general in the paper. Response: The sentence is revised (New line 308-309). Coarse units are more frequent in the upper zone, which make

the average K is much larger in this zone than that in the lower part of the alluvial fan. 22) Lines 293 – 296. It depends. Are you considering arithmetic or geometric average? Response: The sentences were revised. Arithmetic average of hydraulic conductivity was calculated.

Please also note the supplement to this comment:
http://www.hydrol-earth-syst-sci-discuss.net/hess-2016-373/hess-2016-373-AC2-supplement.pdf

---

## Author Comment (AC3) · 25 Oct 2016

When reading the review from anonymous reviewer #3, it clearly arises that he did not like our contribution at all. He did not find even a single merit and just provided a long sequence of extremely negative judgments. For example, "claim . . . unfounded", "previously published", "not original", "not scientific progress", "lacks scientific completeness", "consumed with mundane documentation", "not constitute good science", "important qualities are lacking", and others. This made us astonished, also in the light of the generally positive comments provided by reviewers #1 and #2. Why such a tremendously severe evaluation? Except for a couple of specific issues given in comment 3 "Presentation Quality", all the other comments by reviewer #3 are extremely

general and no suggestion is provided on how the work can be improved.

We strongly disagree with this reviewer and with this way of reviewing a manuscript. Hence, a detail rebuttal of the majority of reviewer #3 comments is given in the following.

1) Scientific Significance - Poor

Rev.#3 : Concepts, ideas, and methods are not new. The claim of an "original method" by the authors is unfounded. Every method used has been previously published and implemented: Dividing a domain into zones to do geostatistical modeling is not original; use of geophysical data to derive facies or hydraulic parameters is not original; assumptions of "K distributions are local stationary" and computing the log10(K) semi-variogram are decades-old concepts. The paper generally reads like documentation of a work assignment, not scientific progress.

Response: This comment is a mixture of obviousness and general statements but the supposed lack of originality is not supported, for example, by specific references. We are obviously conscious that semivariogram, geophysical modelling to infer hydraulic parameters, etc. have been used since a long time. We do not claim originality for this. What is original is the way we integrate a large number of inexpensive and fast VES surveys (properly calibrated through a few more-detail investigations as well logs) with a facies model developed from borehole lithologic data to simulate the log10(K) continuous distribution in multiple-zone heterogeneous alluvial megafans. Moreover, also the application site is anything but worthless, as the Chaobai fan is used to supply the majority of the potable water to one of the most important city in the World. We have specified more clearly the novelty of our contribution in the abstract (lines 16-19) and Introduction (67-71).

2) Scientific Quality - Poor

Rev.#3 (a): In judging scientific quality, consider the scientific method: systematic observation, measurement, and experiment, and the formulation, testing, and modification of hypotheses. Granted, the paper does implement some systematic observation and measurement and proceeds to set up an "experiment" of sorts by producing geostatistical realizations of hydraulic conductivity. However, it is not clear at all what the hypotheses of the paper are and what the "experiment" will actually be: A flow model or a transport model for what use? A calibration/validation exercise to what observations? The paper simply lacks scientific completeness in formulation and testing of hypotheses. The authors seem to be advocating that an "original method" constitutes science or, perhaps, a hypothesis of being "original" is science, but even that is not necessarily true especially since the authors' claim of being "original" is debatable. The closest statement to a hypothesis could find is given at the end of section 2.1: "The characterization of the distribution and spatial variability of hydraulic conductivity is vital for an optimal use of the limited water resources in this area." This statement isn't new or "original" except perhaps at the particular area of study in China. More importantly, this hypothesis is not tested in the paper! Instead, the paper is consumed with mundane documentation of its observations and methods and preparation of an experiment that is never executed. The paper could have tested whether its methods are actually "vital for an optimal use of the limited water resources in this area" (e.g. by flow or transport modeling with comparison to water level or chemistry data, i.e. observations). A scientific result would be proof that the author's methods are better than a typical effective-K model for determination of some "vital" information about the aquifer system. Perhaps the authors plan to do this in another paper, but that does not matter. The existing paper just does not constitute good science on its own.

Response: The major part of this comment is soaked with philosophical argumentations. For example, "The authors seem to be advocating that an "original method" constitutes science", or "the paper is consumed with mundane documentation of its observations and methods", and also "The paper could have tested whether its methods are actually vital for an optimal use of the limited water resources in this area". We think that a reply to such a type of comments is worthlessness. The only two technical

aspects detectable in this part of the review are "what the hypotheses of the paper are" and "what the experiment will actually be". About the former, we have clearly stated in the abstract, introduction, and section 2.4.3 that local stationarity of log10(k) is the basic hypothesis of or investigation. The changes of the variance characterizing the composited semivariogram between the three zones support the utilization of the local-stationary assumption (lines 298-305). Concerning the latter, we have reported in the conclusive section how the outcome of this study will be used in the future: "This result provides valuable insights for understanding the spatial variations of hydraulic conductivity and setting-up groundwater flow, transport, and land subsidence models in alluvial fans" (lines 357-359). As hypothesized by the reviewer, we have planned performing these dynamic simulations in a next step. As supported by reviewers #1 and #2, we believe that the definition of the static model of a complex heterogeneous megafan as that of the Chaobai river is worth to be published.

Rev.#3 (b): From a hydrogeologic perspective, important qualities are lacking in the representation of alluvial fan heterogeneity: (1) there is no directional non-stationarity (e.g. no radial variability of the depositional major axis; no stratigraphic dip), (2) there are abrupt, unrealistic discontinuities between zone (e.g. facies occurrences abruptly terminate and the edge of a zone, like a fault), and (3) the zonal approach leads to unrealistic transitions in geometrical properties (e.g. thickness of gravel deposits).

Response: These are the only scientific inquiries in this review. Curiously, they are the same comments provided by Reviewer #1 in "General comments" #3 and #6. The reviewer can refer to the responses provided to Reviewer #1.

Rev.#3 (c): For all the claims of being an "original method" by combining different methods, the paper does not seem to pay close attention to methods of geology.

Response: Sorry, but we are not able to understand what are the "methods of geology".

3) Presentation Quality - Poor

Rev.#3 (a): Again, the paper is really lacking in actual scientific results (i.e. results of hypothesis testing). The paper is full of documentation of what was done to analyze data and make geostatistical realizations, including re-hashing of old methods with obvious weighting to referencing of the authors' previous publications. Even if the conclusion that "it is worth highlighting we depicted an original method..." were true, this does not constitute good science on its own. The claim of "Fusing multiple-source data" isn't necessarily science on its own since it is routine practice in the earth sciences.

Response: The same response provided to comment 2(a) holds also in this case. What means "The claim of Fusing multiple-source data isn't necessarily science"? We never claimed this! We simply wrote in the Introduction "Recently, data fusion techniques have been developed for coupled inversion of multi-source data to estimate K distributions for groundwater numerical modeling" and add a number of recent references where integration of different data are used to characterize hydrogeological systems. As written above, these are very general suggestions to which it is impossible to provide worthful replies.

Rev.#3 (b): Figures 5 & 6 were never referenced in the text.

Response: The references have been added (lines 268 and 293, respectively).

Rev.#3 (c): Figures 4-6 are difficult to interpret without labeling of y-axis units and use of variable scales in Figure 4 & 5.

Response: The unit of y-axis is already provided in Figure 4 ("Resistivity"), Figures 6 and 7 ("Variance") (previous Figure 4 and 5). Concerning these two latter, we preferred to put the axis units in the first sub-panel and omitted in the others for picture clarity. Concerning the "variable scale" in Figures 6 and 7, in the caption of Figure 6 we already reported "Notice that the range in the y-axis differs for sands and gravel lithology in Zone 2 and Zone 3". The choice is done to improve the clarity of the subpanels. We have updated the caption to "Notice that the range in the y-axis differs for gravel lithology" and add the same note in the caption of Figure 7 too.
Rev.#3 (d): Figure 7 has no scale.

Response: Updated

Rev.#3 (e): These are key elements to geostatistical modeling, yet this information was poorly presented. In terms of documenting what the authors did, the paper is a reasonable piece of communication of the caliber of an institutional report (which would need further revision in regard to Figure 4-7 as noted above and use of English language).

Response: We disagree with the reviewer. We believe that such a negative judgment cannot be based only on the couple of minor figure improvements suggested by this reviewer. Finally, similarly to all the other comments, a negative but extremely general sentence is given for the English form. Any specific example is suggested to improve the English form. Anyway, English language has been updated following the specific recommendations by reviewers #1 and #2.

Please also note the supplement to this comment:
http://www.hydrol-earth-syst-sci-discuss.net/hess-2016-373/hess-2016-373-AC3-supplement.pdf

**Supplement:**

[revised manuscript text omitted]

---

## Author Response (AR1)

Dear prof. Fogg,

We are very grateful for your constructive comments to improve the revision of our manuscript. We carefully revised the text by incorporating the comments one by one.

The detailed revision is presented in the response to each comment:

**Comment 1**. *The authors' response to Comment 2 of Reviewer 1 needs to be reconsidered and reworked because the authors misunderstood what is meant by resistivity log calibration. Because you are using the logged resistivity values quantitatively (e.g., in Archie's law), Reviewer 1 correctly pointed out that to do so requires that the logs have been calibrated. This means that when the logs were run, a geophysical log calibration process should have been done that produces resistivity values that would be consistent enough for one to compare quantitatively the log resistivity values among the different logs. Log calibration is supposed to be done in the field by the person who performs the logging, but in practice, it often is not done and can certainly undermine the validity of Archie's law calculations, among other things. Reviewer 1 also correctly pointed out that salinity of the formation water strongly affects log resistivity, and that too should be taken into account. I do not see any sign that you addressed that.*

*I suspect that you do not know whether the logs were calibrated, and that it is unlikely that you would be able to acquire such information any time soon. Nevertheless, you need to figure out how to modify the manuscript to deal with the log calibration issue. One approach might be to acknowledge that you cannot verify whether the logs were calibrated, and then point out that this presents another source of error that could be reduced in future studies by only using logs that have been calibrated. In essence, you would be working under the assumption that the logs have been calibrated in order to present your work as a proof of concept.*

**Response**: Thank you for the clarification of Comment 2 by Reviewer 1. Yes, we don't know and we cannot verify whether the logs were calibrated or not in the field and how the salinity of the formation water has been accounted for. Following your suggestion, we explicitly acknowledged this lack of information (lines 140-146) and pointed out this might be another source of uncertainty that can be reduced in our future studies (lines 385-386).

*You also need to deal with the issue of groundwater quality (TDS) and it's effects on the resistivity values by either explaining that TDS variations laterally or vertically in the aquifer system are minimal (which seems unlikely) or that this is just another limiting assumption that could be eliminated by performing more complete analyses in the future.*

**Response**: The pore fluid conductivity was estimated by using total dissolved solids (TDS) and temperature data. Lateral and vertical TDS variations in the aquifer system impact on the measured resistivity values. We focused on the resistivity data below water table. Because of the relatively limited dataset and the observed small variability in the TDS, in this paper we ignored the TDS variations in the vertical direction (Line 167-170). In discussion section, we have pointed out that a more complete investigation on the TDS (salinity) distribution in the whole fan will be carried out in the future to improve the reliability of our analyses (lines 382-383).

**Comment 2**. *The authors' response to Reviewer 3 needs some more modification to reflect deeper introspection about their work. I agree that the tone of Reviewer 3's criticisms is somewhat harsh, and from the point of view of an author, can be quite off-putting. There are, however, some important kernels of truth in several of Reviewer 3's comments that are worthy of more careful consideration. In particular, your response to Reviewer 3's Comment 1 needs some reconsideration, given that your assertion of the novelty of the work is based on the incorporation of resistivity logs (see my point 1 above with respect to Reviewer 1). I think that your assertion that the novelty stems from incorporation of the resistivity logs quantitatively into the work may be OK, but here again, you need the caveat(s) regarding how quantitatively representative the logs actually are.*

**Response**: Yes, we believe that the third reviewer also gave us some valuable comments and we address them along with those from Reviewer 1. In our responses to the comments of Reviewer 1 we pointed out that in this study we assumed the logs have been calibrated, which might be another source of uncertainty that can be reduced in future studies.

*Your response to Reviewer 3's statement: "(1) there is no directional non-stationarity (e.g. no radial variability of the depositional major axis; no stratigraphic dip),…" seems incomplete, and I am not sure you fully understood it. Reviewer 3's statement (1) above is basically asking why you did not model the spatial variations in the strike and dip of the fan facies. I am not finding an answer to that question in your responses to either Reviewer 1 (whom you refer to in this context) or to Reviewer 3. It appears to me that you did not have information on variations in dip or strike orientations of the variogram structure, which is why you did not account for that. This sort of thing should be included in the part of your paper that discusses model limitations and how it might be improved in the future.*

**Response**: In our previous response we stated that they were the same comments provided by Reviewer #1 in "General comments" #6. The reviewer could refer to the responses provided to Reviewer #1. Now we added more clarification to these comments as the editor suggested. We have used available information along the dip direction, see Figure 7. In the description of Figure 9, we added a sentence to indicate that "since we simulate the dip direction along the main water flow direction and the strike-directional semivariogram is assumed to be similar as that in dip direction (due to lack of enough data to estimate the parameters for the strike-directional semivariogram), the simulated facies in the fan apex do not show a radiating pattern. More information about simulating the radiating pattern can be found from Carle and Fogg (1997) and Fogg et al. (1998)" (Line 330-334).

The estimated range and variance for the semivariograms of dip direction in Zone 2 and Zone 3 were given in Table 4 and Fig. 7. Accurate descriptions of the semivarigrams in the dip and lateral directions will be included in our future study to improve the developed three-dimensional permeability field (Line 383-385).

The list of all relevant changes made in the manuscript was as follows.

1) We added some sentences (Line 140-146) on logs calibration according to the editor's suggestion.
2) Detail information on simulating the stochastic faices were given from Line 266 to 270.
3) We added more clarification on semivariogram (Line 330-332) as the editor suggested.

[revised manuscript text omitted]

Zone1

Zone 2

Zone 3

**Figure 7**

[Figure]

Zone 2

[Figure]

Zone 3

**Figure 8**

[Figure]

Zone 1 Zone 2 Zone 3

**Figure 9**

[Figure]